# Modified rational six vertex model on the rectangular lattice

**Samuel Belliard[1*], Rodrigo Alves Pimenta[2†] and Nikita A. Slavnov[3‡]**

**1** Institut Denis-Poisson, Université de Tours, Université d'Orléans,
Parc de Grammont, 37200 Tours, France
**2** Department of Physics and Astronomy, University of Manitoba, Winnipeg R3T 2N2, Canada
**3** Steklov Mathematical Institute of Russian Academy of Sciences,
8 Gubkina str., Moscow, 119991, Russia

⋆ samuel.belliard@lmpt.univ-tours.fr , † rodrigo.alvespimenta@umanitoba.ca ,
‡ nslavnov@mi-ras.ru

## Abstract

We consider a rational six vertex model on a rectangular lattice with boundary conditions that generalize the usual domain wall type. We find that the partition function of the inhomogeneous version of this model is given by a modified Izergin determinant. The proofs are based on the quantum inverse scattering method and its representation theory together with elementary linear algebra.

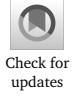

# 1  Introduction

A partition function is the central object in Statistical Mechanics. Computing the partition function is often a difficult combinatorial problem that sometimes can be solved exactly, particularly in two dimensional lattices [1–3]. A prominent example is the six vertex model, which can be defined on a rectangular lattice by assigning two possible states to each of its edges (see Figure 1). The possible configurations around a vertex are constrained by the ice rule. Boundary conditions must be imposed, and it turns out that they play an important role in the nature of the mathematical representation of the partition function.

Different solvable boundary conditions have been considered for the six vertex model. The most common are the torus, the cylinder and fixed boundary conditions like domain wall and reflecting end, as well as mixtures of them. The first two types can be computed via the diagonalization of an appropriate transfer matrix, which can in principle be done using Bethe ansatz, as early noted by Lieb in his solution of the square ice [4]. On the other hand, fixed boundary conditions like domain wall type admit a determinant representation [5, 6], so do the reflecting end case [7]. In this paper, we will focus on fixed boundary conditions that generalize the domain wall type.

The domain wall boundary condition for the six vertex model in the square lattice was introduced by Korepin [5], where a recurrence relation for the partition function was found. The solution of the recurrence was later given by Izergin [6] in the form of a determinant (see, for example, the monograph [8] for details). The partition function on the rectangular lattice, also called a partial domain wall boundary, was considered recently and it is also given by a determinant [9].

Here we propose a generalization of the rational six vertex partition function on the rectangular lattice with four arbitrary boundaries. In the language of the quantum inverse scattering method, this partition function is given by certain expectation values of a string of modified creation operators, which arise in the context of the modified algebraic Bethe ansatz (see for instance [10–14] and references therein). The expectation values are associated with four arbitrary wall states labeled with the compass directions (30). We argue that the partition function satisfies a homogeneous system of linear equations, which follows directly from the (modified) triangular representation theory of the Yangian $Y(gl(2))$ and Yang-Baxter algebra via certain off-shell relations. We then show that the linear system is solved by a modified Izergin determinant (89).

Let us recall that the partition function of the six vertex model with domain wall boundary condition was shown to solve a system of functional equations [15] (see also [16]), as well as a system of algebraic equations [17].

The linear system approach was recently discovered in the computation of scalar products between on-shell and off-shell Bethe states [18]. Its powerfulness has been demonstrated in the computation of on-shell/off-shell scalar products of Bethe states in the closed XYZ chain [19] and in the open XXZ chain with general integrable boundaries [20]. Here we show that it can also be used to compute partition functions opening a new avenue of possibilities.

This paper is organized as follows. In Section 2, we recall the definition of an arbitrary vertex model on the rectangular lattice, its partition function with arbitrary twists and some essential ingredients of the quantum inverse scattering method. In Section 3, auxiliary operators and the associated representation theory are studied in the rational six vertex case. Next, in Section 4, we derive the homogeneous linear systems satisfied by the partition function and construct the solution in the form of a modified Izergin determinant. Our concluding remarks and further directions of research are presented in Section 5. Some technical details are given in the appendices A and B.

**Notation.** We use a shorthand notation for sets of variables and products over them. For example, we denote a set of $m$ variables $u_j$ by $\bar{u} = \{u_1, \ldots, u_m\}$. We usually leave the cardinality implicit and note it as $\#\bar{u} = m$. The removal of the $i$-th element of the set $\bar{u}$ is denoted $\bar{u}_i = \bar{u} \backslash u_i$. For the product of two variable function $g(u, v)$ over the set $\bar{u}$ we use,

$$g(z, \bar{u}) = \prod_{x \in \bar{u}} g(z, x), \quad g(\bar{u}, z) = \prod_{x \in \bar{u}} g(x, z), \quad g(\bar{u}, \bar{v}) = \prod_{x \in \bar{u}, y \in \bar{v}} g(x, y). \tag{1}$$

We will also use such notation for the product of commuting operators

$$B(\bar{u}) = \prod_{x \in \bar{u}} B(x). \tag{2}$$

If no product is involved, a vertical bar is used to indicate the multivariable function dependency, *e.g.*,

$$s(u|\bar{u}) = s(u|u_1, \ldots, u_m), \quad r(\bar{u}|\bar{w}) = r(u_1, \ldots, u_m|v_1, \ldots, v_n). \tag{3}$$

The following functions will be used,

$$g(u, v) = \frac{c}{u - v}, \quad f(u, v) = \frac{u - v + c}{u - v}, \quad h(u, v) = \frac{u - v + c}{c}, \quad \tilde{h}(u, v) = \frac{u - v - c}{c}. \tag{4}$$

They satisfy the relations

$$f(u, v) = g(u, v) + 1 = g(u, v)h(u, v) = g(v, u)\tilde{h}(v, u). \tag{5}$$

## 2 Partition function and quantum groups

In this section, we review some basic concepts in the theory of integrable vertex models, including the operator formulation of the partition function with general open boundary condition. The material in this section is valid for an arbitrary integrable vertex model with arbitrary number of states.

### 2.1 R-matrix and Yang-Baxter equation

Let us recall the inhomogeneous vertex model on the $m \times n$ rectangular lattice from the quantum group formalism (see Figure 1). A finite dimensional vector space denoted $V_{a_i}$ carrying a free parameter $u_i$, also called inhomogeneity, is associated with each line of the lattice $i = 1, \ldots, m$. Similarly, a finite dimensional vector space $V_{b_i}$ with parameter $v_i$ is associated with each column $i = 1, \ldots, n$. Every vertex is labeled by a pair of variables $(u, v)$ and it is encoded by an invertible matrix $R_{ab}(u, v)$ (see Figure 2). The entries of this matrix are the statistical weights of the model. Explicitly, we have,

$$R_{ab}(u, v) = \sum_{jk, \ell m} R_{jk, \ell m}(u, v)(E_{jk})_a \otimes (E_{\ell m})_b, \tag{6}$$

where the $(E_{ij})_a$ are standard basis matrices (that satisfy $E_{ij}E_{kl} = \delta_{jk}E_{il}$) acting on the space $V_a$, and the sum is taken over $(j, k) \in \{1, \ldots, \dim(V_a)\}^2$ and $(\ell, m) \in \{1, \ldots, \dim(V_b)\}^2$. One imposes that this matrix is a solution of the Yang Baxter equation,

$$R_{ab}(u, v)R_{ac}(u, w)R_{bc}(v, w) = R_{bc}(v, w)R_{ac}(u, w)R_{ab}(u, v), \tag{7}$$

and therefore ensures the integrability of the model.

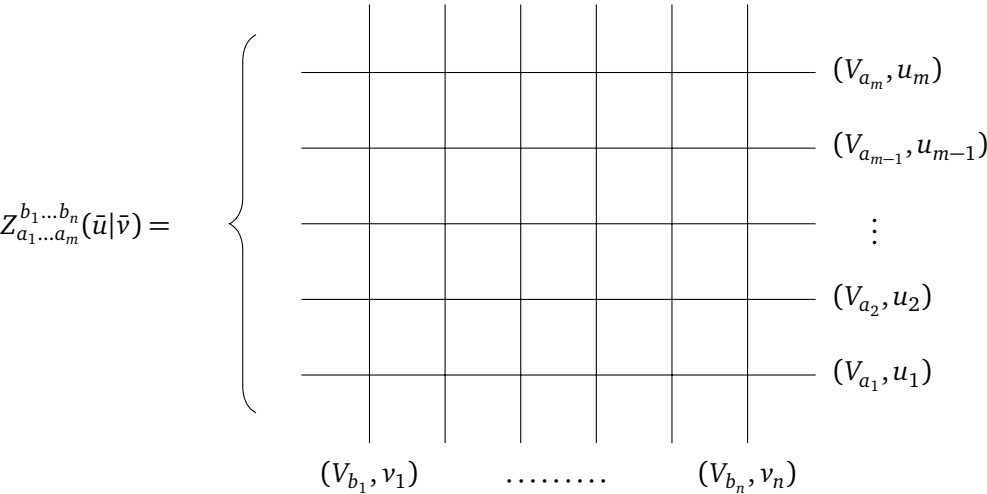

$$Z^{b_1...b_n}_{a_1...a_m}(\bar{u}|\bar{v}) =$$

Figure 1: Inhomogeneous vertex model on the $m \times n$ rectangular lattice.

$$R_{ab}(u,v) = \quad (V_a, u) \quad\rule{}{}\quad$$

$$(V_b, v)$$

Figure 2: Graphical picture of a vertex and the R-matrix.

In the so-called auxiliary space formalism, we can write the matrix of the partition function in the form,

$$Z^{b_1...b_n}_{a_1...a_m} = \overrightarrow{\prod_{i=1}^{m}}\;\overrightarrow{\prod_{j=1}^{n}}R_{a_i b_j}(u_i, v_j)\,, \tag{8}$$

which is an endomorphism of $V_{a_1} \otimes \cdots \otimes V_{a_m} \otimes V_{b_1} \otimes \cdots \otimes V_{b_n}$. The following exchange relations follow from the Yang-Baxter algebra,

$$R_{a_i a_{i+1}}(u_i, u_{i+1})Z^{b_1...b_n}_{a_1...a_i a_{i+1}...a_m} = Z^{b_1...b_n}_{a_1...a_{i+1} a_i...a_m}R_{a_i a_{i+1}}(u_i, u_{i+1})\,, \tag{9}$$

and

$$R_{b_i b_{i+1}}(v_i, v_{i+1})Z^{b_1...b_i b_{i+1}...b_n}_{a_1...a_m} = Z^{b_1...b_{i+1} b_i...b_n}_{a_1...a_m}R_{b_i b_{i+1}}(v_i, v_{i+1})\,. \tag{10}$$

We now assume that the R-matrix has the properties,

$$[R_{ab}(u,v), B_a B_b] = [R_{ab}(u,v), \hat{B}_a \hat{B}_b] = 0\,, \tag{11}$$

where $B$ and $\hat{B}$ are matrices in $\text{End}(V)$. Using the matrices $B$ and $\hat{B}$ we can add a twist to each auxiliary space. This motivates the definition of a "quasiperiodic", or twisted inhomogeneous partition function, by taking the trace over each space, namely,

$$Z_{mn}(\bar{u}|\bar{v}|B|\overline{B}) = \text{tr}_{\bar{a},\bar{b}}\left( (\prod_{i=1}^{n}\hat{B}_{b_i})(\prod_{j=1}^{m}B_{a_j})Z^{b_1...b_n}_{a_1...a_m}(\bar{u}|\bar{v}) \right)\,, \tag{12}$$

where we used the shorthand notation (1,3). This definition generalizes the standard definition of the periodic partition function given by $B_a = \hat{B}_a = 1_{V_a}$, that always satisfies (11)

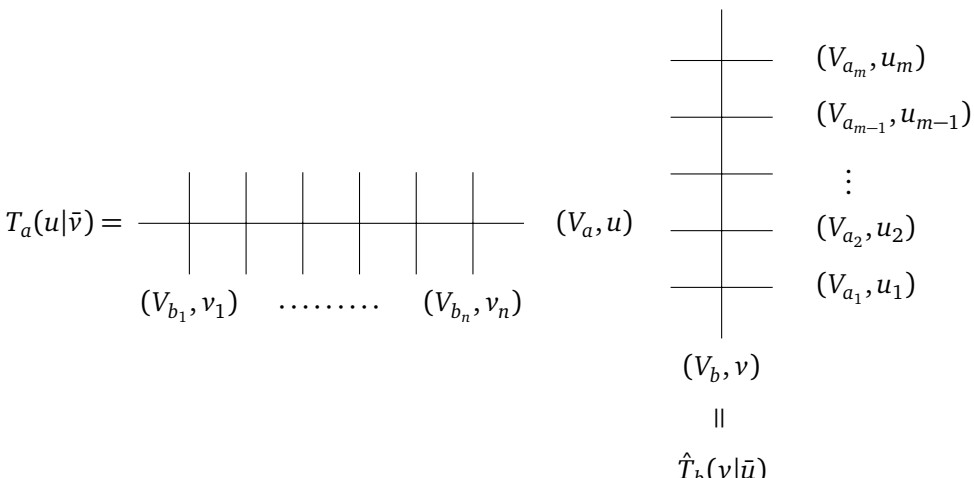

Figure 3: Single row monodromy matrix (left) and single column monodromy matrix (right).

(see *e.g.* Chapter 7.5 of [21]). It is easy to show that (12) is a symmetric function over the sets $\bar{u}$ or $\bar{v}$. Moreover, if we relax the symmetric property for one set of variables, we can introduce different twist matrices for each vector space. That is, the transformation $\prod_{i=1}^{n} \hat{B}_{b_i} \rightarrow \prod_{i=1}^{n} \hat{B}_{b_i}^{(i)}$ breaks the symmetry over permutation of the set $\bar{v}$ and the transformation $\prod_{j=1}^{m} B_{a_j} \rightarrow \prod_{j=1}^{m} B_{a_j}^{(j)}$ breaks the symmetry over permutation of the set $\bar{u}$.

## 2.2 Operator formalism and quantum group

We now express the matrix of the partition function (8) in terms of products of single row monodromy matrices, either in the horizontal or vertical direction of the lattice.

The horizontal single row monodromy matrix is given by

$$T_a(u|\bar{v}) = \overrightarrow{\prod_{j=1}^{n}} R_{ab_j}(u, v_j), \tag{13}$$

and can be represented as in the left panel of Figure 3. Is is clear that (8) can be written as,

$$Z_{a_1 \ldots a_m}^{b_1 \ldots b_n}(\bar{u}|\bar{v}) = \overrightarrow{\prod_{i=1}^{m}} T_{a_i}(u_i|\bar{v}). \tag{14}$$

The single row monodromy matrix (13) satisfies the RTT relation,

$$R_{ab}(u_a, u_b) T_a(u_a|\bar{v}) T_b(u_b|\bar{v}) = T_b(u_b|\bar{v}) T_a(u_a|\bar{v}) R_{ab}(u_a, u_b). \tag{15}$$

It allows one to define the transfer matrix with an arbitrary twist $B_a$ given by

$$B(u) = \operatorname*{tr}_a(B_a T_a(u|\bar{v})), \tag{16}$$

which is represented on the left panel of Figure 4, and it is integrable since it leads to a family of mutually commuting operators,

$$[B(u_1), B(u_2)] = 0. \tag{17}$$

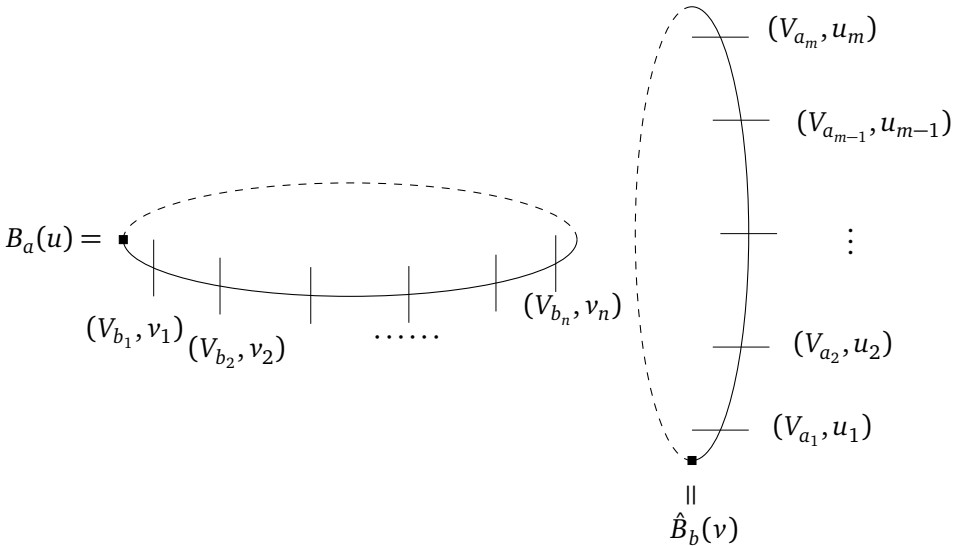

Figure 4: Modified operators $B(u)$ (left) and $\hat{B}(v)$ (right). The twists are represented by the filled squares.

The partition function can then be written in the operator formulation,

$$Z_{mn}(\bar{u}|\bar{v}|B|\overline{B}) = \operatorname*{tr}_{\bar{b}}\Big(\big(\prod_{i=1}^{n}\hat{B}_{b_i}\big)B(\bar{u})\Big). \tag{18}$$

Similarly, we can consider the single column monodromy matrix defined by,

$$\hat{T}_b(v|\bar{u}) = \overrightarrow{\prod_{i=1}^{m}} R_{a_i b}(u_i, v) \tag{19}$$

(see right panel of Figure 3). It satisfies the RTT relation,

$$R_{ba}(v_b, v_a)\hat{T}_a(v_a|\bar{u})\hat{T}_b(v_b|\bar{u}) = \hat{T}_b(v_b|\bar{u})\hat{T}_a(v_a|\bar{u})R_{ba}(v_b, v_a), \tag{20}$$

from which the additional family of mutually commuting operators

$$\hat{B}(v) = \operatorname*{tr}_{b}(\hat{B}_b \hat{T}_b(v|\bar{u})), \tag{21}$$

with arbitrary twist $\hat{B}_b$ can be obtained, by tracing over a given auxiliary space $V_b$, (See a graphical representation in the right panel of Figure 4). A second reformulation of the matrix of partition function follows from $\hat{T}$, namely,

$$Z_{a_1\ldots a_m}^{b_1\ldots b_n}(\bar{u}|\bar{v}) = \overrightarrow{\prod_{i=1}^{n}} \hat{T}_{b_i}(v_i|\bar{u}), \tag{22}$$

and leads to the partition function in terms of the transfer matrix $\hat{B}(u)$,

$$Z_{mn}(\bar{u}|\bar{v}|B|\overline{B}) = \operatorname*{tr}_{\bar{a}}\Big(\big(\prod_{i=1}^{m}B_{a_i}\big)\hat{B}(\bar{v})\Big). \tag{23}$$

We have therefore two equivalent transfer matrix formulations (18) and (23) of the partition function (12). Depending on the nature of the twists $\{B, \hat{B}\}$, we have different boundary

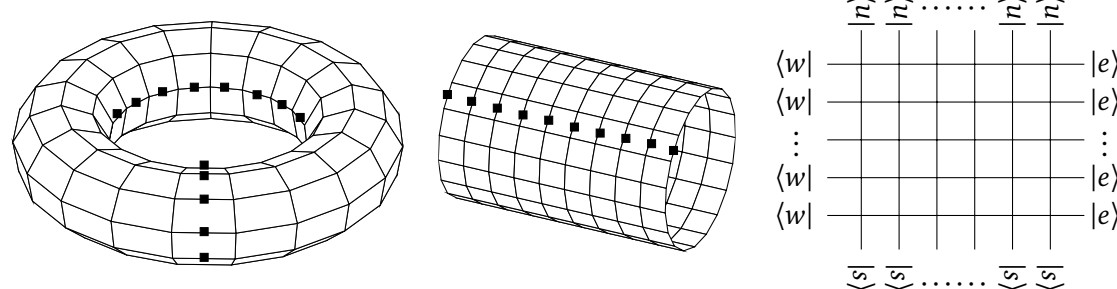

Figure 5: Partition function in different geometries according to the nature of the twist.

conditions for the partition function (see Figure 5). If both twists do not have rank one, we have a torus type of partition function. If only one of the twists has rank one, we have a cylinder and if both satisfy this condition we have the plane. If one or both of the twists have rank one, the trace reduces to matrix elements of the matrices $B(\bar{u})$ or $\hat{B}(\bar{v})$ (see next (29)).

Indeed, let us suppose that $\text{rank}(B) = \text{rank}(\hat{B}) = 1$ and therefore that $\det(B) = \det(\hat{B}) = 0$. We can then find the following bi-vector formulation,

$$B = |e\rangle \otimes \langle w| = \sum_{ij} w_i e_j E_{ij}, \qquad \hat{B} = |n\rangle \otimes \langle s| = \sum_{ij} n_i s_j E_{ij}, \qquad (24)$$

where

$$|x\rangle = \sum_i x_i |i\rangle, \quad \langle x| = \sum_i x_i \langle i|, \qquad (25)$$

for arbitrary labels $x \in \{w, e, n, s\}$ and we use the relation $|i\rangle \otimes \langle j| = E_{ij}$ with $|i\rangle$ the vector with 1 at the row $i$ and zero elsewhere, $\langle j|$ is its dual and $\langle j|i\rangle = \delta_{ij}$. It allows to rewrite operators (16) and (21) as,

$$B(u) = \text{tr}_a(B_a T_a(u|\bar{v})) = {}_a\langle w|T_a(u|\bar{v})|e\rangle_a = \sum_{ij} w_i e_j t_{ij}(u), \qquad (26)$$

$$\hat{B}(v) = \text{tr}_b(\hat{B}_b \hat{T}_b(v|\bar{u})) = {}_b\langle s|\hat{T}_b(v|\bar{u})|n\rangle_b = \sum_{ij} s_i n_j \hat{t}_{ij}(v). \qquad (27)$$

We recall that operators of this type arise in the modified Bethe ansatz [10], and can be seen as a null transfer matrix as $\det(B) = \det(\hat{B}) = 0$ [11]. They are represented in Figure 6.

Note that we have the trace identity

$$\text{tr}_a(\hat{B}_a X_a) = {}_a\langle s|X_a|n\rangle_a, \qquad (28)$$

for any matrix $X$ acting on vector space $V_a$. Then, the general partition function can be rewritten into the forms of expectation values of product of operators,

$$Z_{mn}(\bar{u}|\bar{v}|B|\hat{B}) = \langle S|B(\bar{u})|N\rangle = \langle W|\hat{B}(\bar{v})|E\rangle, \qquad (29)$$

with

$$\begin{aligned} \langle W| &= \langle w| \otimes \cdots \otimes \langle w|, & |E\rangle &= |e\rangle \otimes \cdots \otimes |e\rangle, \\ \langle S| &= \langle s| \otimes \cdots \otimes \langle s|, & |N\rangle &= |n\rangle \otimes \cdots \otimes |n\rangle, \end{aligned} \qquad (30)$$

which can be interpreted as a partition function for the six vertex model with four arbitrary walls. The form (29) is suitable to be treated within the quantum inverse scattering method and linear algebra. For the full open case both forms are possible and there is no quantization and no Bethe equations. We will see in the next section the rational six vertex case and its determinant representations.

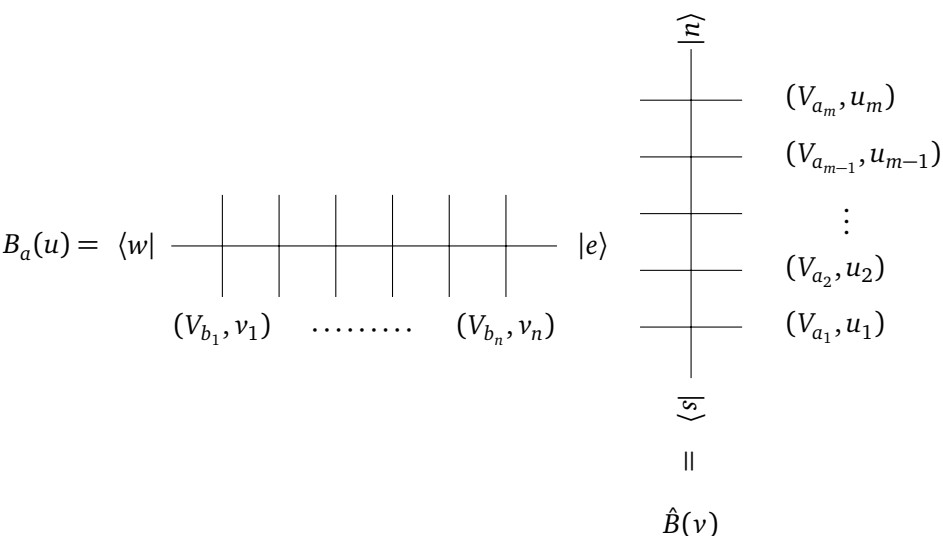

Figure 6: Modified operators when $\text{rank}(B) = \text{rank}(\hat{B}) = 1$.

$$
u \;\;\; \overset{j}{\underset{j}{\;\;\big|\;\;}}\;\; {}^{j}_{\;} \overset{j}{} = \frac{u-v+c}{c} \qquad\qquad u \;\;\; \overset{k}{\underset{j}{\;\;\big|\;\;}} {}^{k} = \frac{u-v}{c} \qquad\qquad u \;\;\; \overset{k}{\underset{j}{\;\;\big|\;\;}} {}^{k} = 1
$$

$a$-type vertex $\qquad\qquad\qquad$ $b$-type vertex $\qquad\qquad\qquad$ $c$-type vertex

Figure 7: The nonzero Boltzmann weights $a$, $b$ and $c$ of the six vertex model.

## 3 Rational six vertex model case: auxiliary operators and triangular representation theory

All considerations so far are valid for vertex models having an arbitrary number of states per edge. For concreteness, in the following we only consider the symmetric (zero field) six vertex model, with 2 states in all edges of the lattice and therefore $V_{a_i} = V_{b_j} = \mathbb{C}^2$. The three possible nonzero Boltzmann weights, called type $a$, $b$ and $c$, are represented in Figure 7. The corresponding R-matrix is given by

$$
R_{ab}(u-v) = \frac{u-v}{c} I_{ab} + P_{ab}, \tag{31}
$$

with $I_{ab}$ the identity operator and $P_{ab}$ the permutation operator on $\mathbb{C}^2 \otimes \mathbb{C}^2$. It is one of the simplest solution of the Yang-Baxter equation (7) and can be used to define the Yangian $Y(gl_2)$, as we briefly recall in Appendix A.

Let us introduce additional operators $A(u)$ and $D(u)$. First consider the two zero determinant 2 by 2 matrices in bi-vector form, where the notations are adopted as in (25),

$$
A = |a\rangle \otimes \langle \tilde{a}| = \sum_{ij} \tilde{a}_i a_j E_{ij}, \tag{32}
$$

$$
D = |d\rangle \otimes \langle \tilde{d}| = \sum_{ij} \tilde{d}_i d_j E_{ij}. \tag{33}
$$

They satisfy $\det(A) = \det(D) = 0$ by definition. Then, we have the operators,

$$A(u) = \operatorname*{tr}_a(A_a T_a(u)) = \langle \tilde{a} | T_a(u) | a \rangle = \sum_{ij} a_j \tilde{a}_i t_{ij}(u), \tag{34}$$

$$D(u) = \operatorname*{tr}_a(D_a T_a(u)) = \langle \tilde{d} | T_a(u) | d \rangle = \sum_{ij} d_j \tilde{d}_i t_{ij}(u). \tag{35}$$

Imposing the relations

$$B_a A_b P_{ab} = B_a A_b, \tag{36}$$

$$P_{ab} B_a D_b = B_a D_b, \tag{37}$$

which are equivalent to the constraints $\tilde{a}_j = w_j$ and $d_j = e_j$, where normalizations were absorbed in the free parameters $a_j$ and $\tilde{d}_j$, we obtain the following exchange relations,

$$A(u)B(v) = f(v,u)B(v)A(u) + g(u,v)B(u)A(v), \tag{38}$$

$$D(u)B(v) = f(u,v)B(v)D(u) + g(v,u)B(u)D(v). \tag{39}$$

They directly follow from the definition of $\{A(u), D(u), B(u)\}$ as linear combinations of the Yangian generators $t_{ij}(u)$ (see appendix A), and can also be calculated directly from (A.7) multiplying by $A_a B_b$, taking the traces over the spaces $V_a$ and $V_b$, and finally using relations (36).

From linear combination of the results of Theorem A.1, we can find the action of these operators on the general states (30). We find the following action of such operators on the general states (30),

$$A(u)|N\rangle = a_N \lambda_1(u)|N\rangle + c_N B(u)|N\rangle, \tag{40}$$

$$D(u)|N\rangle = d_N \lambda_2(u)|N\rangle + f_N B(u)|N\rangle, \tag{41}$$

with

$$a_N = \frac{\langle a|\sigma_y|e\rangle \langle w|n\rangle}{\langle n|\sigma_y|e\rangle}, \quad c_N = \frac{\langle a|\sigma_y|n\rangle}{\langle e|\sigma_y|n\rangle}, \quad d_N = \frac{\langle e|\sigma_y|n\rangle \langle w|\sigma_y|\tilde{d}\rangle}{\langle w|n\rangle}, \quad f_N = \frac{\langle \tilde{d}|n\rangle}{\langle w|n\rangle}, \tag{42}$$

and

$$\lambda_1(u_j) = h(u_j, \bar{v}), \quad \lambda_2(u_j) = \frac{1}{g(u_j, \bar{v})}. \tag{43}$$

Note that we have the relations

$$a_N + \operatorname{tr}(B)c_N = \operatorname{tr}(A), \quad d_N + \operatorname{tr}(B)f_N = \operatorname{tr}(D). \tag{44}$$

Similarly, the action of such operators on the general dual states (30) is given by,

$$\langle S|A(u) = a_S \lambda_2(u)\langle S| + c_S \langle S|B(u),$$

$$\langle S|D(u) = d_S \lambda_1(u)\langle S| + f_S \langle S|B(u), \tag{45}$$

with

$$a_S = \frac{\langle a|\sigma_y|e\rangle \langle s|\sigma_y|w\rangle}{\langle s|e\rangle}, \quad c_S = \frac{\langle s|a\rangle}{\langle s|e\rangle}, \quad d_S = \frac{\langle s|e\rangle \langle \tilde{d}|\sigma_y|w\rangle}{\langle s|\sigma_y|w\rangle}, \quad f_S = \frac{\langle s|\sigma_y|\tilde{d}\rangle}{\langle s|\sigma_y|w\rangle}, \tag{46}$$

and the relations

$$a_S + \text{tr}(B)c_S = \text{tr}(A), \quad d_S + \text{tr}(B)f_S = \text{tr}(D). \tag{47}$$

For further convenience, we introduce the parameter $\beta$ given by,

$$\beta = \frac{a_S}{a_N} = \frac{d_N}{d_S} = \frac{\langle w|\sigma_y|s\rangle\langle e|\sigma_y|n\rangle}{\langle e|s\rangle\langle w|n\rangle} = -\frac{\text{tr}(B\sigma_y\hat{B}^t\sigma_y)}{\text{tr}(B\hat{B})}, \tag{48}$$

that also satisfies,

$$\text{tr}(B\sigma_y\hat{B}^t\sigma_y) + \text{tr}(B\hat{B}) = \text{tr}(B)\,\text{tr}(\hat{B}). \tag{49}$$

## 4 Linear system and modified Izergin determinant formula

In this section, we derive a linear system for the partition function following the method proposed in [18]. The first step is to note that the exchange relations (38) imply the following multiple actions,

$$A(u_i)B(\bar{u}_i) = \sum_{j=1}^{m+1} \frac{f(\bar{u}_j, u_j)}{h(u_i, u_j)} B(\bar{u}_j)A(u_j), \tag{50}$$

$$D(u_i)B(\bar{u}_i) = \sum_{j=1}^{m+1} \frac{f(u_j, \bar{u}_j)}{h(u_j, u_i)} B(\bar{u}_j)D(u_j), \tag{51}$$

for a set $\bar{u} = \{u_1, \ldots, u_{m+1}\}$ which includes an extra parameter $u_{m+1}$.

Next, we act with (50) on the general states $\{|N\rangle, \langle S|\}$. For convenience, consider the quantities,

$$F_A = \langle S|(A(u_i) - c_S B(u_i))B(\bar{u}_i)|N\rangle, \tag{52}$$

$$F_D = \langle S|(D(u_i) - f_S B(u_i))B(\bar{u}_i)|N\rangle, \tag{53}$$

and compute their actions to the left and to the right taking into account (45,40). We obtain,

$$\frac{\text{tr}(B)}{\chi} \sum_{j=1}^{m+1} \left( -\beta\lambda_2(u_i)\delta_{ij} + \lambda_1(u_j)\frac{f(\bar{u}_j, u_j)}{h(u_i, u_j)} \right) Z_{mn}(\bar{u}_j|\bar{v}|B|\hat{B}) = Z_{m+1n}(\bar{u}|\bar{v}|B|\hat{B}), \tag{54}$$

$$\frac{\text{tr}(B)}{\chi} \sum_{j=1}^{m+1} \left( \lambda_1(u_i)\delta_{ij} - \beta\lambda_2(u_j)\frac{f(u_j, \bar{u}_j)}{h(u_j, u_i)} \right) Z_{mn}(\bar{u}_i|\bar{v}|B|\hat{B}) = Z_{m+1n}(\bar{u}|\bar{v}|B|\hat{B}), \tag{55}$$

where

$$\chi = 1 - \beta = \frac{\text{tr}(\hat{B})\,\text{tr}(B)}{\text{tr}(B\hat{B})}. \tag{56}$$

In the case $m = n$ the following theorem was proven in [11],

**Theorem 4.1.**

$$Z_{n+1n}(\bar{u}|\bar{v}|B|\hat{B}) = \text{tr}(B) \sum_{j=1}^{n+1} g(u_j, \bar{u}_j)\lambda_1(u_j)\lambda_2(u_j)Z_{nn}(\bar{u}_j|\bar{v}|B|\hat{B}). \tag{57}$$

Now, define a vector $\vec{X} = (X_1, \ldots, X_{m+1})^t$ with $X_j = Z_{nn}(\bar{u}_j|\bar{v}|B|\hat{B})$. Using (57) in (54,55) for $m = n$, we obtain the following homogeneous linear systems,

$$L_A \vec{X} = 0, \qquad L_D \vec{X} = 0, \tag{58}$$

where the matrices $L_A, L_D$ with dimension $(n+1) \times (n+1)$ have entries given by

$$(L_A)_{ij} = -\beta \lambda_2(u_j)\delta_{ij} + g(u_j, \bar{u}_j)Y_A(u_j|\bar{u}_i), \tag{59}$$

$$(L_D)_{ij} = \lambda_1(u_j)\delta_{ij} - g(u_j, \bar{u}_j)Y_D(u_j|\bar{u}_i), \tag{60}$$

with

$$Y_A(u_j|\bar{u}_i) = \tilde{h}(u_j, \bar{u}_i)\lambda_1(u_j) - \chi \lambda_1(u_j)\lambda_2(u_j), \tag{61}$$

$$Y_D(u_j|\bar{u}_i) = \beta h(u_j, \bar{u}_i)\lambda_2(u_j) + \chi \lambda_1(u_j)\lambda_2(u_j), \tag{62}$$

where we used (44,47) for the constants.

Each homogeneous system in (58) has a nontrivial solution if $\det(L_{A,D}) = 0$ which implies that $\mathrm{rank}(L_{A,D}) \leq n$. To prove that the determinants vanish, we define the nonsingular $(n+1) \times (n+1)$ matrix $W$ with entries,

$$W_{ij} = \frac{g(u_j, \bar{u}_j)}{g(u_j, \bar{w}_i)}, \tag{63}$$

where a new set of arbitrary pairwise distinct parameters $\bar{w} = \{w_1, \ldots, w_{n+1}\}$, that will be specified later, is introduced. The determinant of $W$ is given by the ratio of determinants of the Vandermonde type,

$$\det(W) = \frac{\Delta(\bar{w})}{\Delta(\bar{u})}, \qquad \Delta(\bar{u}) = (\prod_{i<j} g(u_i, u_j))^{-1}. \tag{64}$$

Now, we define the products $\tilde{L}_A = W L_A$ and $\tilde{L}_D = W L_D$. Using the relations,

$$\sum_{k=1}^{n+1} W_{ik} = 1, \quad \sum_{k=1}^{n+1} W_{ik}\tilde{h}(u_j, \bar{u}_k) = \tilde{h}(u_j, \bar{w}_i), \quad \sum_{k=1}^{n+1} W_{ik}h(u_j, \bar{u}_k) = h(u_j, \bar{w}_i), \tag{65}$$

that can be proven using contour integral for appropriate rational functions (see [22]), we find,

$$(\tilde{L}_A)_{ij} = g(u_j, \bar{u}_j)\left(-\beta \frac{\lambda_2(u_j)}{g(u_j, \bar{w}_i)} + \lambda_1(u_j)\tilde{h}(u_j, \bar{w}_i) - \chi \lambda_1(u_j)\lambda_2(u_j)\right), \tag{66}$$

$$(\tilde{L}_D)_{ij} = g(u_j, \bar{u}_j)\left(\frac{\lambda_1(u_j)}{g(u_j, \bar{w}_i)} - \beta \lambda_2(u_j)h(u_j, \bar{w}_i) - \chi \lambda_1(u_j)\lambda_2(u_j)\right). \tag{67}$$

Let $i = n+1$. We set $w_j = v_j - c$ for $j \neq n+1$ in the $A$ system and $w_j = v_j$ for $j \neq n+1$ in the $D$ system. We also define $w_{n+1} = w$. Then, we find, respectively,

$$(\tilde{L}_A)_{n+1j}|_{w_j=v_j-c, w_{n+1}=w} = g(u_j, \bar{u}_j)\lambda_1(u_j)\lambda_2(u_j)(1-\beta-\chi) = 0, \tag{68}$$

$$(\tilde{L}_D)_{n+1j}|_{w_j=v_j, w_{n+1}=w} = g(u_j, \bar{u}_j)\lambda_1(u_j)\lambda_2(u_j)(1-\beta-\chi) = 0, \tag{69}$$

from which it follows that $\det(L_A) = \det(L_D) = 0$. For the other rows $1 \leq i \leq n$, we have

$$(\tilde{L}_A)_{ij} = \tilde{h}(v_i, w)g(u_j, \bar{u}_j)\lambda_1(u_j)\lambda_2(u_j)\left(-\beta g(u_j, v_i - c) + g(u_j, v_i)\right), \tag{70}$$

$$(\tilde{L}_D)_{ij} = \frac{g(u_j, \bar{u}_j)}{g(v_i, w)}\lambda_1(u_j)\lambda_2(u_j)\left(-\beta g(u_j, v_i - c) + g(u_j, v_i)\right). \tag{71}$$

After renormalization with appropriate diagonal matrices,

$$(\tilde{T}_A)_{ij} = \frac{\delta_{ij}}{\tilde{h}(v_i, w)}, \quad (\tilde{T}_D)_{ij} = g(v_i, w)\delta_{ij}, \tag{72}$$

we find the same transformed linear system for both $A$ and $D$ systems, namely,

$$(\tilde{L})_{ij} = g(u_j, \bar{u}_j)\lambda_1(u_j)\lambda_2(u_j)\left(-\frac{\beta}{h(u_j, v_i)} + g(u_j, v_i)\right). \tag{73}$$

Then, from Cramer's solution of the homogeneous linear system and relations (43), it follows that,

$$Z_{nn}(\bar{u}|\bar{v}|B|\hat{B}) = \phi(\bar{v})\frac{\det_n(M)}{\det_n(C)}. \tag{74}$$

Here $\phi(\bar{v})$ is some symmetric function in $\bar{v}$, and

$$M_{ij} = (\tilde{L})_{ij}^{(n+1)}\frac{g(u_j, \bar{v})}{g(u_j, \bar{u}_j)} = h(u_j, \bar{v})\left(-\beta\frac{1}{h(u_j, v_i)} + g(u_j, v_i)\right), \tag{75}$$

are the entries of the reduced matrix $(\tilde{L})_{ij}^{(n+1)}$ corresponding to the matrix $\tilde{L}$ with the $n+1$ row and column removed. We also introduced a Cauchy matrix with elements

$$C_{ij} = g(u_i, v_j), \tag{76}$$

that have the well known determinant,

$$\det_n(C) = g(\bar{u}, \bar{v})\Delta(\bar{u})\Delta'(\bar{v}). \tag{77}$$

Here $\Delta'(\bar{u}) = (\prod_{i>j} g(u_i, u_j))^{-1}$ which is similar to $\Delta(\bar{u})$ in (64).

We finally need to fix the normalization in $\bar{v}$. This is done by comparing the asymptotic limit $\bar{u} \to \infty$ from the determinant formula (74) and from the definition of the partition function (29). The later follows simply from the asymptotic of the $B(u)$ operator due to the R-matrix (31),

$$\lim_{u \to \infty} B(u) = \left(\frac{u}{c}\right)^n \mathrm{tr}(B) + \cdots, \tag{78}$$

which leads to,

$$\lim_{\bar{u} \to \infty}\left(Z_{nn}(\bar{u}|\bar{v}|B|\hat{B})/(\bar{u}/c^n)^n\right) = (\mathrm{tr}(\hat{B})\,\mathrm{tr}(B))^n + \cdots \tag{79}$$

On the other hand, we have to take the asymptotic limit on the determinant form (74). To do that we use the inverse of the Cauchy determinant, see *e.g.* [22], given by,

$$C_{kl}^{-1} = g(u_l, v_k)\frac{g(\bar{v}_k, v_k)g(u_l, \bar{u}_l)}{g(\bar{u}, v_k)g(u_l, \bar{v})}, \tag{80}$$

that satisfies the following summation rules,

$$\sum_{l=1}^{n} C_{il}^{-1} g(u_l, v_j) = \delta_{ij}, \tag{81}$$

and

$$\sum_{l=1}^{n} C_{il}^{-1}\frac{1}{h(u_l, v_j)} = \frac{g(v_i, \bar{v}_i)\tilde{h}(v_j, \bar{v}_j)}{g(\bar{u}, v_i)h(\bar{u}, v_j)h(v_i, v_j)}. \tag{82}$$

It follows that (74) can be put into the form,

$$Z_{nn}(\bar{u}|\bar{v}|B|\hat{B}) = \frac{\phi(\bar{v})}{g(\bar{u},\bar{v})} \det_n \left( -\beta \delta_{ij} + \frac{f(\bar{u},v_i)f(v_i,\bar{v}_i)}{h(v_i,v_j)} \right). \tag{83}$$

Then, we can take the infinite limit for the set $\bar{u}$ and find

$$\lim_{\bar{u}\to\infty} Z/(\bar{u}/c^n)^n = \phi(\bar{v}) \det_n \left( -\beta \delta_{ij} + \frac{f(v_j,\bar{v}_j)}{h(v_i,v_j)} \right) + \cdots \tag{84}$$

It follows from (B.8) and Lemma 4.1 in [23], namely, from the identity

$$\sum f(\bar{u}_{\mathrm{i}}, \bar{u}_{\mathrm{ii}}) = \binom{n}{p}, \tag{85}$$

where the sum is taken over partition of the set $\bar{u}$ into two sets $\{\bar{u}_{\mathrm{i}}, \bar{u}_{\mathrm{ii}}\}$ with fixed size $\bar{u}_{\mathrm{i}} = p$, $\bar{u}_{\mathrm{ii}} = n - p$. From Newton binomial formula it follows

$$\lim_{\bar{u}\to\infty} Z/(\bar{u}/c^n)^n = \phi(\bar{v})\chi^n + \cdots \tag{86}$$

Comparing (86) with (79) we find,

$$\phi(\bar{v}) = \left( \frac{\mathrm{tr}(\hat{B})\,\mathrm{tr}(B)}{\chi} \right)^n, \tag{87}$$

that leads to

$$Z_{nn}(\bar{u}|\bar{v}|B|\hat{B}) = \frac{\mathrm{tr}(B)^n\,\mathrm{tr}(\hat{B})^n}{\chi^n} \lambda_2(\bar{u}) K_{nn}^{(\beta)}(\bar{u}|\bar{v}), \tag{88}$$

where $K_{nn}^{(\beta)}$ is the modified Izergin determinant (B.1) for $m = n$, with $\lambda_2(\bar{u}) = (g(\bar{u},\bar{v}))^{-1}$.

Finally, using the limits of the modified Izergin determinant in Proposition (B.1), the formula (88) can be extended for values of $m \neq n$. For $m > n$, we use the limit (B.5) to eliminate one $v_j$, while for $n > m$ we use the limit (B.6) one $u_j$. Comparing with operator form (29), we can find the general case $m, n$,

$$Z_{mn}(\bar{u}|\bar{v}|B|\hat{B}) = \frac{\mathrm{tr}(B)^m\,\mathrm{tr}(\hat{B})^n}{\chi^n} \lambda_2(\bar{u}) K_{mn}^{(\beta)}(\bar{u}|\bar{v}), \tag{89}$$

that can be expressed in two possible forms,

$$Z_{mn}(\bar{u}|\bar{v}|B|\hat{B}) = \frac{\mathrm{tr}(B)^m\,\mathrm{tr}(\hat{B})^n}{\chi^n} \lambda_2(\bar{u}) \det_n \left( -\beta \delta_{jk} + \frac{f(\bar{u},v_j)f(v_j,\bar{v}_j)}{h(v_j,v_k)} \right), \tag{90}$$

and

$$Z_{mn}(\bar{u}|\bar{v}|B|\hat{B}) = \frac{\mathrm{tr}(B)^m\,\mathrm{tr}(\hat{B})^n}{\chi^m} \lambda_2(\bar{u}) \det_m \left( \delta_{jk} f(u_j,\bar{v}) - \beta \frac{f(u_j,\bar{u}_j)}{h(u_j,u_k)} \right). \tag{91}$$

Furthermore, due to the properties of the modified Izergin determinant in Proposition (B.2), we have the additional presentations as sum over partitions of the set $\bar{v}$,

$$Z_{mn}(\bar{u}|\bar{v}|B|\hat{B}) = \frac{\mathrm{tr}(B)^m\,\mathrm{tr}(\hat{B})^n}{\chi^n} \lambda_2(\bar{u}) \sum_{\bar{v}\Rightarrow\{\bar{v}_{\mathrm{I}},\bar{v}_{\mathrm{II}}\}} (-\beta)^{\#\bar{v}_{\mathrm{II}}} f(\bar{u},\bar{v}_{\mathrm{I}}) f(\bar{v}_{\mathrm{I}},\bar{v}_{\mathrm{II}}), \tag{92}$$

or over the set $\bar{u}$,

$$Z_{mn}(\bar{u}|\bar{v}|B|\hat{B}) = \frac{\mathrm{tr}(B)^m\,\mathrm{tr}(\hat{B})^n}{\chi^m} \lambda_2(\bar{u}) \sum_{\bar{u}\Rightarrow\{\bar{u}_{\mathrm{I}},\bar{u}_{\mathrm{II}}\}} (-\beta)^{\#\bar{u}_{\mathrm{I}}} f(\bar{u}_{\mathrm{II}},\bar{v}) f(\bar{u}_{\mathrm{I}},\bar{u}_{\mathrm{II}}). \tag{93}$$

## 5 Conclusion

Using the triangular representation theory of the modified operators (26,34) together with the exchange relations (50), we found linear systems that characterize, up to normalization, the partition function of the inhomogeneous rational six vertex model on the rectangular lattice with two arbitrary twists $\{B, \hat{B}\}$ of rank $\text{rank}(B) = \text{rank}(\hat{B}) = 1$. Due to symmetry, only one of the eight generic boundary parameters associated with the compass states (30) is free, that is, the partition function can be written solely in terms of the parameter $\beta$, up to normalization by $\text{tr}(B)^m \text{tr}(\hat{B})^n$.

The solution of the linear system and, therefore, the partition function on the rectangular lattice is given by a modified Izergin determinant (89). For specific values of the parameters, one can recover known examples in the literature, for example, the partial domain wall partition function [9] or the Izergin-Korepin determinant ($m = n$, $s_1 = 0, s_2 = 1, e_1 = 0, e_2 = 1$ and $n_1 = 1, n_2 = 0, w_1 = 1, w_2 = 0$) [5,6].

For future problems, it is interesting to consider the linear system approach to the computation of the partition function of the six vertex model with reflecting boundary conditions, related to the reflection algebra [24], which has been extensively studied [7, 25–29]. Also in this context, the linear system approach may find applications in the connection between certain scalar products of modified Bethe states and q-Racah polynomials [30].

Another intriguing problem is to study the trigonometric six vertex model (associated with the XXZ chain) under anti-periodic boundary conditions. The first step here would be to find appropriate modified operators and off-shell relations, which are not yet known for this model, despite recent developments (see [31–33] and references therein).

It is also important to investigate higher rank vertex models, initially those based on the $sl(n)$ algebra [34]. In this case, the off-shell relations and, therefore, the associated linear systems are more intricate (see [35–37]).

## Acknowledgments

S.B. thanks F. Levkovich-Maslyuk, R. Frassek, J. Jacobsen, V. Pasquier, D. Serban, V. Terras for discussions. R.A.P. thanks R. Frassek for discussions

**Funding information**    R.A.P. acknowledges support by the German Research Council (DFG) via the Research Unit FOR 2316.

## A  Basics about the Yangian $Y(gl_2)$

In this appendix, we review some basic facts about the Yangian of $gl_2$ denoted $Y(gl_2)$ (see *e.g.* the monograph [21,38] for details), as formulated in the quantum inverse scattering method. Let us introduce the monodromy matrix

$$T(u) = \begin{pmatrix} t_{11}(u) & t_{12}(u) \\ t_{21}(u) & t_{22}(u) \end{pmatrix}, \tag{A.1}$$

whose elements are given by the formal series in $u$,

$$t_{ij}(u) = \delta_{ij} + \sum_{r=1}^{\infty} t_{ij}^{(r)} u^{-r}, \tag{A.2}$$

where $t_{ij}^{(r)}$ are the generators of the Yangian $Y(gl_2)$ subject to the defining RTT relations

$$R_{ab}(u-v)T_a(u)T_b(v) = T_b(v)T_a(u)R_{ab}(u-v), \tag{A.3}$$

where $R$ is the six vertex R-matrix (31). They encode exchange relations for the elements $t_{ij}(u)$, for instance,

$$t_{11}(u)t_{12}(v) = f(v,u)t_{12}(v)t_{11}(u) + g(u,v)t_{12}(u)t_{11}(v), \tag{A.4}$$

$$t_{22}(u)t_{12}(v) = f(u,v)t_{12}(v)t_{22}(u) + g(v,u)t_{12}(u)t_{22}(v), \tag{A.5}$$

$$t_{12}(u)t_{12}(v) = t_{12}(v)t_{12}(u), \tag{A.6}$$

where the rational functions $f, g$ in the formal variables $u, v$ are given by (4).

Let us note that (A.3) can be rewritten as

$$[T_a(u), T_b(v)] = g(v,u)P_{ab}(T_a(u)T_b(v) - T_a(v)T_b(u)). \tag{A.7}$$

We consider finite dimensional representations of the Yangian [39] from (13). Highest and lowest weight representations can be respectively constructed from vectors $|0\rangle$ and $|\hat{0}\rangle$ with actions,

$$T_a(u)|0\rangle = \begin{pmatrix} \lambda_1(u) & t_{12}(u) \\ 0 & \lambda_2(u) \end{pmatrix}_a |0\rangle, \tag{A.8}$$

$$T_a(u)|\hat{0}\rangle = \begin{pmatrix} \lambda_2(u) & 0 \\ t_{21}(u) & \lambda_1(u) \end{pmatrix}_a |\hat{0}\rangle. \tag{A.9}$$

The dual analogs $\langle 0|$ and $\langle \hat{0}|$ have actions,

$$\langle 0|T_a(u) = \langle 0| \begin{pmatrix} \lambda_1(u) & 0 \\ t_{21}(u) & \lambda_2(u) \end{pmatrix}_a, \tag{A.10}$$

$$\langle \hat{0}|T_a(u) = \langle \hat{0}| \begin{pmatrix} \lambda_2(u) & t_{12}(u) \\ 0 & \lambda_1(u) \end{pmatrix}_a. \tag{A.11}$$

In particular, for the 6 vertex case, we have the highest and lowest vectors,

$$\begin{aligned}
|0\rangle &= \otimes_{j=1}^n \begin{pmatrix} 1 \\ 0 \end{pmatrix}, & \langle 0| &= \otimes_{j=1}^n \begin{pmatrix} 1, & 0 \end{pmatrix}, \\
|\hat{0}\rangle &= \otimes_{j=1}^n \begin{pmatrix} 0 \\ 1 \end{pmatrix}, & \langle \hat{0}| &= \otimes_{j=1}^n \begin{pmatrix} 0, & 1 \end{pmatrix},
\end{aligned} \tag{A.12}$$

and the weight functions $\lambda_i(u)$ are given by (43).

The following theorem provides the modified version of the actions (A.8,A.9,A.10,A.11).

**Theorem A.1.** *The action of the entries of the monodromy matrix on an arbitrary vector*

$$|X\rangle = |x\rangle \otimes \cdots \otimes |x\rangle, \quad with \quad |x\rangle = \begin{pmatrix} x_1 \\ x_2 \end{pmatrix}, \tag{A.13}$$

*with $x_1 \neq 0$ are given by*

$$t_{11}(u)|X\rangle = \lambda_1(u)|X\rangle - \frac{x_2}{x_1}t_{12}(u)|X\rangle, \tag{A.14}$$

$$t_{22}(u)|X\rangle = \lambda_2(u)|X\rangle + \frac{x_2}{x_1}t_{12}(u)|X\rangle, \tag{A.15}$$

$$t_{21}(u)|X\rangle = \frac{x_2}{x_1}(\lambda_1(u) - \lambda_2(u))|X\rangle - \left(\frac{x_2}{x_1}\right)^2 t_{12}(u)|X\rangle, \tag{A.16}$$

*and with $x_2 \neq 0$ are given by*

$$t_{11}(u)|X\rangle = \lambda_2(u)|X\rangle + \frac{x_1}{x_2}t_{21}(u)|X\rangle, \tag{A.17}$$

$$t_{22}(u)|X\rangle = \lambda_1(u)|X\rangle - \frac{x_1}{x_2}t_{21}(u)|X\rangle, \tag{A.18}$$

$$t_{12}(u)|X\rangle = \frac{x_1}{x_2}(\lambda_1(u) - \lambda_2(u))|X\rangle - \left(\frac{x_1}{x_2}\right)^2 t_{21}(u)|X\rangle. \tag{A.19}$$

*Similarly, we can find the action on the dual vector*

$$\langle X| = \langle x| \otimes \cdots \otimes \langle x|, \quad with \quad \langle x| = (\begin{array}{cc} x_1, & x_2 \end{array}), \tag{A.20}$$

*with $x_1 \neq 0$*

$$\langle X|t_{11}(u) = \lambda_1(u)\langle X| - \frac{x_2}{x_1}\langle X|t_{21}(u), \tag{A.21}$$

$$\langle X|t_{22}(u) = \lambda_2(u)\langle X| + \frac{x_2}{x_1}\langle X|t_{21}(u), \tag{A.22}$$

$$\langle X|t_{12}(u) = \frac{x_2}{x_1}(\lambda_1(u) - \lambda_2(u))\langle X| - \left(\frac{x_2}{x_1}\right)^2 \langle X|t_{21}(u), \tag{A.23}$$

*with $x_2 \neq 0$*

$$\langle X|t_{11}(u) = \lambda_2(u)\langle X| + \frac{x_1}{x_2}\langle X|t_{12}(u), \tag{A.24}$$

$$\langle X|t_{22}(u) = \lambda_1(u)\langle X| - \frac{x_1}{x_2}\langle X|t_{12}(u), \tag{A.25}$$

$$\langle X|t_{21}(u) = \frac{x_1}{x_2}(\lambda_1(u) - \lambda_2(u))\langle X| - \left(\frac{x_1}{x_2}\right)^2 \langle X|t_{12}(u). \tag{A.26}$$

*Proof.* These relations can be proven in the following way. Let us introduce an invertible matrix $X$ such that $X|0\rangle = |x\rangle$ with the form,

$$X = \left( \begin{array}{cc} x_1 & \alpha \\ x_2 & \beta \end{array} \right). \tag{A.27}$$

Then due to the $gl(2)$ invariance of the Yangian we have $[T_a(u), X_a \prod_{i=1}^{n} X_{b_i}] = 0$ and it follows that,

$$T_a(u)|X\rangle = T_a(u)\left(\prod_{i=1}^{n} X_{b_i}\right)|0\rangle = T_a(u)X_a\left(\prod_{i=1}^{n} X_{b_i}\right)(X_a)^{-1}|0\rangle \tag{A.28}$$

$$= X_a\left(\prod_{i=1}^{n} X_{b_i}\right)T_a(u)|0\rangle(X_a)^{-1}. \tag{A.29}$$

From the representation theory of the Yangian (A.8, A.9), we have,

$$T_a(u)|0\rangle = \begin{pmatrix} \lambda_1(u) & 0 \\ 0 & \lambda_2(u) \end{pmatrix}_a |0\rangle + \begin{pmatrix} 0 & 1 \\ 0 & 0 \end{pmatrix}_a t_{12}(u)|0\rangle. \tag{A.30}$$

It follows that,

$$T_a(u)|X\rangle = \Lambda_a(u)|X\rangle + E_a\Big(\prod_{i=1}^{N} X_{b_i}\Big)t_{12}(u)|0\rangle, \tag{A.31}$$

with

$$\det(X)\Lambda_a(u) = X_a \begin{pmatrix} \lambda_1(u) & 0 \\ 0 & \lambda_2(u) \end{pmatrix}_a (X_a)^{-1} \tag{A.32}$$

$$= \begin{pmatrix} x_1\beta\lambda_1(u) - x_2\alpha\lambda_2(u) & -x_1\alpha\big(\lambda_1(u) - \lambda_2(u)\big) \\ x_2\beta\big(\lambda_1(u) - \lambda_2(u)\big) & x_1\beta\lambda_2(u) - x_2\alpha\lambda_1(u) \end{pmatrix}_a, \tag{A.33}$$

and

$$\det(X)E_a = X_a \begin{pmatrix} 0 & 1 \\ 0 & 0 \end{pmatrix}_a (X_a)^{-1} = \begin{pmatrix} -x_1 x_2 & (x_1)^2 \\ -(x_2)^2 & x_1 x_2 \end{pmatrix}_a. \tag{A.34}$$

Then after some linear algebra we express $\big(\prod_{i=1}^{N} X_{b_i}\big)t_{12}(u)|0\rangle$ in terms of $t_{12}(u)|X\rangle$ and $|X\rangle$ we find the desired actions. The left actions can be proven similarly and we omit it here.

$\square$

# B  Modified Izergin determinant

We recall some basic properties of the modified Izergin determinant (see [12, 13] for more details). The modified Izergin determinant can be defined as follows.

**Definition B.1.** *Let $\bar{u} = \{u_1, \ldots, u_m\}$, $\bar{v} = \{v_1, \ldots, v_n\}$ and $z$ be a complex number. Then the modified Izergin determinant $K_{mn}^{(z)}(\bar{u}|\bar{v})$ is defined by*

$$K_{mn}^{(z)}(\bar{u}|\bar{v}) = \det_n\left(-z\delta_{jk} + \frac{f(\bar{u}, v_j)f(v_j, \bar{v}_j)}{h(v_j, v_k)}\right). \tag{B.1}$$

*Alternatively the modified Izergin determinant can be presented as*

$$K_{mn}^{(z)}(\bar{u}|\bar{v}) = (1-z)^{n-m} \det_m\left(\delta_{jk}f(u_j, \bar{v}) - z\frac{f(u_j, \bar{u}_j)}{h(u_j, u_k)}\right). \tag{B.2}$$

The proof of the equivalence of representations (B.1) and (B.2) can be found in [40].

In the particular case $z = 1$ and $\#\bar{u} = \#\bar{v} = n$, the modified Izergin determinant turns into the ordinary Izergin determinant, that we traditionally denote by $K_n(\bar{u}|\bar{v})$,

$$K_{nn}^{(1)}(\bar{u}|\bar{v}) = K_n(\bar{u}|\bar{v}). \tag{B.3}$$

It also follows from (B.2) that

$$K_{mn}^{(1)}(\bar{u}|\bar{v}) = 0, \quad \text{for} \quad m < n. \tag{B.4}$$

Additional properties of the modified Izergin determinant needed here are given in the following propositions.

**Proposition B.1.** *We have the limits,*

$$\lim_{v_j \to \infty} K^{(z)}_{mn}(\bar{u}|\bar{v}) = (1-z)K^{(z)}_{mn-1}(\bar{u}|\bar{v}_j), \tag{B.5}$$

$$\lim_{u_j \to \infty} K^{(z)}_{mn}(\bar{u}|\bar{v}) = K^{(z)}_{m-1n}(\bar{u}_j|\bar{v}). \tag{B.6}$$

**Proposition B.2.** *We have the sum formulation,*

$$K^{(z)}_{mn}(\bar{u}|\bar{v}) = \sum_{\bar{v} \Rightarrow \{\bar{v}_{\mathrm{I}}, \bar{v}_{\mathrm{II}}\}} (-z)^{\#\bar{v}_{\mathrm{II}}} f(\bar{u}, \bar{v}_{\mathrm{I}}) f(\bar{v}_{\mathrm{I}}, \bar{v}_{\mathrm{II}}), \tag{B.7}$$

*where the sum is taken over all partitions $\bar{v} \Rightarrow \{\bar{v}_{\mathrm{I}}, \bar{v}_{\mathrm{II}}\}$, and*

$$K^{(z)}_{m,n}(\bar{u}|\bar{v}) = (1-z)^{n-m} \sum_{\bar{u} \Rightarrow \{\bar{u}_{\mathrm{I}}, \bar{u}_{\mathrm{II}}\}} (-z)^{\#\bar{u}_{\mathrm{I}}} f(\bar{u}_{\mathrm{II}}, \bar{v}) f(\bar{u}_{\mathrm{I}}, \bar{u}_{\mathrm{II}}), \tag{B.8}$$

*where the sum is taken over all partitions $\bar{u} \Rightarrow \{\bar{u}_{\mathrm{I}}, \bar{u}_{\mathrm{II}}\}$.*

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
