# Peer review of "Modified rational six vertex model on the rectangular lattice"

_SciPost Physics, doi:SciPost Phys. 16, 009 (2024)_

## Round 1 · Referee Report · Anonymous (Referee 1) · 2023-10-25

Strengths

There is a new result in this paper. It is a good new result, which will have interesting developments

Weaknesses

Did not any abnormal weaknesses.

Report

I recommend to publish it as is

Requested changes

No changes requested

---

## Round 1 · Referee Report · Anonymous (Referee 2) · 2023-10-28

Strengths

Solid and original result in the are of exactly solvable model .

Weaknesses

No weaknesses

Report

The paper consider the rational six-vertex model on a rectangular
domain of the square lattice, with fixed boundary conditions that
generalize the known "domain wall" and "partial domain wall" ones
(for the square and rectangular domain, respectively). More precisely,
while in such cases boundary edges were set to a state of spin up or
down, here they are fixed to a generic vector in $\mathbb{C}^2$.

The inhomogeneous version of the model is considered, and the
corresponding partition function is evaluated in the framework of the
Quantum Inverse Scattering Method. AS a result, two different
`modified Izergin determinant' representation are provided.

The subject is rather technical, and the paper addresses a very
specialized audience. This said, the paper is extremely well written
and clear, and of great interest.

The paper provides a remarkable extension of previous fundamental
results in the theory of quantum integrable models. It also open the
way to further developments on the subject.

As a side comment concerning possible further developments, let me add
to those mentioned by the authors in their conclsions: i) the
homogeneous limit of the partition function, and ii) its asymptotic
behaviour in the scaling/thermodynamic limit.

In conclusion, the paper fulfills completely all necessary
requirements of originality, scientific rigour, relevance, clarity,
and interest.

I strongly recommend publication, modulo corrections of a few typos,
see below.

Requested changes

1) after eq (1.1) : verb is missing; 2) last sentence of Sect. 2.1: the last formula should be rewritten as $\prod_{i=1}^m B_{a_i} \to \prod_{i=1}^m B_{a_i} ^{(i)}$ ; 3) eq (2.14) : the product should run over index $i$ ; 4) 3rd line of Sect. 3: erase "for"; 5) line before (3.8): "the" -> "we"; 6) after eq. (4.4): "the their" -> "their"; 7) eq. (4.15) and (4.28): the notation $\Delta(\bar{u})$ is a bit confusing, since it essentially coincides to the inverse of a Vandermonde determinant. But I leave to the author the decision about a possible improvement. 8) eq. (4.43) : the sum should be over $\bar{v}\Rightarrow { \bar{v}{\mathrm{I}},\bar{v}}$} ( instead of $\bar{u}_{\mathrm{II}}$).

---

## Round 1 · Referee Report · Anonymous (Referee 3) · 2023-11-27

Strengths

New and important generalisation of the Izergin determinant formula for the partition function of six-vertex model with fixed boundary conditions
Clear and detailed presentation of the result and proofs.

Weaknesses

None

Report

Determinant representation for the partition function of the six vertex with domain wall boundary conditions (Izergin formula) can be considered as the most crucial element for the computation of correlation functions and form factors of the spin chains. In this paper the authors derive a very interesting generalisation of this formula for a very general class of fixed boundary conditions. The final results have a very simple form and very probably will be extremely useful in particular for the study of open spin chains with non-diagonal boundaries.
The paper is well written and the results are derived in a very clear way.

Requested changes

Some proofreading to eliminate several typos (such as repeated words)

---

## Round 2 · Author Response

We thank the referees for their positive opinion on our paper and for the time and effort they took to review it. We have corrected several typos noted by the referees. For clarification, we added a comment before (4.15) clarifying that $\Delta(\bar u)$ is given by a ratio of determinants of the Vandermonde type.
Sincerely,
The Author
Sincerely,
The Author

---

## Editorial Decision

published